# A study of consumer-generated advertising—An experimental analysis based on purchase experiences, information channels, and privacy costs

Chuanxi Cai ✱, Yu Liu

School of Economic and Management, Nanjing Forestry University, Nanjing, Jiangsu, China

✱ ccx@njfu.edu.cn

## Abstract

Taking consumer-generated advertising as the research object, hypotheses are proposed on the influential relationship between purchase experiences, information channels and privacy costs and consumer-generated advertising. Then, a between-groups experimental design was adopted. For 402 samples, through questionnaires and interview experiments, ANOVA was used to analyze consumer-generated advertising, exploring the relationship between the behavior of consumer-generated advertising and their influence factors in different consumption scenarios. The results of the study show that: (1)information channels, purchase experiences and privacy costs all have significant effects on consumer-generated advertising. (2)The effect of purchase experiences on consumer-generated advertising is moderated by information channels. (3)The effect of privacy costs on consumer-generated advertising is moderated by information channels and purchase experiences. (4)There is no significant interaction effect among purchase experiences, information channels and privacy costs. The results of the study are helpful for enterprises to manage consumer-generated advertising reasonably and enhance the marketing management capabilities of enterprises.

## 1 Introduction

With the growing popularity of social media and online platforms, more and more users create, share and publish content on them on their own, i.e., user-generated content (UGC) [1], such as reviews, photos, videos and blogs. At the same time, more and more people tend to trust the information provided by consumers rather than that provided by enterprises. And 93% of consumers trust online user-generated content more than branded content [2]. As a result, social media and online platforms, as an important way to reach consumers, have contributed to content generated by consumers, gradually becoming a major influence factor for consumers

**Data availability statement:** All relevant data are within the paper and its Supporting Information files.

**Funding:** This work was supported in part by The Science and Technology Innovation Fund (163060171) and in part by the General Program in philosophy and Social Sciences (2022SJYB0124).

**Competing interests:** The authors have declared that no competing interests exist.

purchases. In this context, enterprises have begun to combine word-of-mouth recommendations with advertising, resulting in consumer-generated advertising [3] has gradually become an important form of generating advertising. Consumer-generated advertising primarily involves enterprises incentivizing consumers through material rewards or user experience, prompting them to voluntarily share their products or brands to social media platforms like WeChat Moments, forward to microblogging, or participate in platforms such as Dianping. This fosters consumer-consumer interaction, so as to generate and promote their advertisements with the help of consumers' efforts.

In addition, consumer-generated advertising, with its characteristics of low cost, high response and easy acceptance, has the potential to become an important way of generating online advertisements in the future. For example, Lululemom encourages both new and existing users to share fitness photos and videos wearing lululemon sportswear by giving away products and membership points, forming a community culture with the theme of "sweaty life". The topic has accumulated more than 1.46 million pieces of content on Instagram [4]. Research shows that consumer-generated advertising centered on user autonomy, where individuals selectively share their personal stories and experiences with friends (e.g., WeChat Monernts can be set to be visible to only a few people). This approach has a relatively low privacy cost, and the ads are high interactive, capable of evoking empathy among friends and enhancing the conversion rate of ads. Data shows that users' trust in consumer-generated advertising (4.25/5 points) is significantly higher than that in enterprise-generated advertising (3.1/5 points) [5], including those generated by artificial intelligence.

From the above cases, it is not difficult to see that, apart from rewards, there are at least three other important factors affecting consumers to generate advertisements. First, **purchase experiences**, that is, consumers' purchase history, may cause them to develop prejudice against a certain brand, thus affecting their trust in this brand and further increasing the possibility of their participation in activities; second, **information channels**, that is, different ways for consumer to obtain a product information will lead to different levels of trust in the product, thus affecting consumers' willingness to generate advertisements; and third, **privacy costs** refer to the private information that may be exposed in consumer-generated advertising, such as personal life status, psychological emotions, and interests and hobbies. These can raise consumers' privacy concerns and, in turn, affect their willingness to generate advertisements.

**Existing studies mainly focused qualitatively on the direct impact and moderating effect of purchase experiences** (Yao Xi and Jian Yufan 2016) [6], **information channels** (Yang Zhou et al. 2017) [7] **and privacy costs** (Xie Yi et al. 2020) [8] **on consumer behavior from the perspective of enterprises, without consideration of consumers themselves. That is to say, they lacked the exploration of the intrinsic influence mechanism and action path that affect consumer's generation of advertisements from the perspective of the causes and internal psychological mechanisms of consumer-generated advertising.** Now, we will conduct an analysis from the perspective of consumers' psychological mechanism.

First, compared with the products acquired by consumers themselves, the products recommended from friends and family are more likely to motivate consumers to generate advertisements. Second, compared with the branded products that have never been purchased, the branded products that have been purchased are more likely to motivate consumers to generate advertisements. Third, some consumers may view sharing advertisements with high privacy cost as a status signal to showcase their tastes or flaunt wealth, and with this need for self-presentation potentially outweighing privacy concerns. Fourth, for consumers who have previously purchased a particular brand, if their purchase experiences was highly positive, they may perceive the potential benefits of sharing ads (such as referral rewards, brand recognition) as offsetting some privacy risks, thereby increasing their willingness to generate ads. Conversely, if their purchase experiences was negative, consumers will be more inclined to protect their privacy and avoid the risk of privacy breaches associated generating ads.

Therefore, beyond privacy costs, purchase experiences and information channels also significantly affect consumer-generated advertising. Based on this, this study examines the interplay among purchase experiences, information channels and privacy costs in relation to consumer-generated advertising. Theoretically, it enriches research on the motivations and influence factors of consumer-generated advertising. Practically, it provides guidance for enterprises to scientifically and reasonably steer consumer-generated advertising. Additionally, this paper analyzes the potential interactions between purchase experiences, information channels and privacy costs, providing marketing insights for enterprises considering whether to utilize and how to manage consumer-generated advertising.

## 2 Literature review and research hypotheses

### 2.1 A study on the motivations and influence factors of consumer-generated advertising

Campbell et al. (2011) [9] thoroughly explored the motivation behind consumer-generated advertising, identifying intrinsic enjoyment, self-improvement, and perspective-shifting as the driving forces behind such creative endeavors. "Intrinsic enjoyment" refers to individuals with artistic inclination deriving pleasure from the creative process itself through the application of specific skills; "Self-enhancement" denotes consumers utilizing ads creation to achieve personal growth and self-promotion; "Perspective-shifting" describes consumers crafting advertisements to affect or alter others' established thought patterns or viewpoints regarding a brand. Meng Lei and Chai Jinping (2013) [10] used an empirical analysis method to investigate the driving factors of consumer-generated advertising from the perspective of consumers' response. Findings revealed that corporate reputation positively affects consumers' trust and attitude, and consumers' identification positively affects consumers' attitude, and both factors positively affect consumers' behavioral intention of generating advertisements. On this basis, many scholars have conducted research on the influence factors of consumer-generated advertising.

Many factors affect consumer-generated advertising and their relationships are complex. By constructing a relevant model, Yao Xi and Jian Yufan (2016) [6] proposed that environment, experience, stimulation and motivation are the four key factors driving consumer-generated advertising behavior within social media, and there are intricate paths among them. Jian Yufang (2016) [11] establish a model of factors hindering consumers' online ad generation behavior, through conducted in-depth interviews with consumers selected from microblogs, WeChat, and snapping. The study conclude that specific online social network connections and atmosphere, related negative experiences, low security factors, and consumers' perceptions of social risk, psychological risk, functional risk, and security risk are the primary factors hindering consumers' online ads generation behavior. It further proposed specific causal pathways for each factor and formulated 16 concrete empirical research hypotheses. Yang Zhou et al. (2017) [7] proposed five categories of motivations for consumer-generated advertising: self-concept perception, self-indulgence, community interaction, empowerment and collective creation. This confirmed the effect of factors such as trust, source credibility, similarity, and advertising nature on consumer responses. Wang et al. (2017) [12] explored the factors affecting the authenticity of consumer-generated advertising from three perspectives: product type, advertising media, and audience gender.

Additionally, consumer-generated advertising, as a consumer referral strategy, is also affected by referral incentives. Yimin Zhu et al. (2011) [13] argue that the incentive levels enhance consumers' willingness to refer, with this willingness increasing progressively as incentives grow. Incentive distribution schemes also affect referral willingness: compared with rewarding only existing customers, consumers exhibit a stronger willingness to recommend when both existing and new customers are rewarded. The word-of-mouth communication channel and product involvement level exert moderating effect on these two influences. On this basis, many scholars have conducted research on recommendation reward strategy (Jiang Fenfen et al. 2020 [14]; Duan Yongrui et al. 2022 [15]) and recommendation willingness (Lu Pingjun et al. 2021 [16]).

With the development of big data and short-from videos platforms like Shakeology, the motivations and influences factors behind consumer-generated advertising are also evolving, which might even involve relationship between consumer's recommendations or consumer's world of mouth through social media and companies that benefit from these recommendations. For example, Xie Yi et al. (2020) [8] analyzed the direct impact and moderating effect of privacy concerns on consumer behavior. Song Bo (2022) [17] found that when confronted with the collection and use of a large amount of privacy information, consumers exhibit two distinct privacy behaviors: on the one hand, concerns about privacy breaches and loss of data ownership further drive them to reject new technology; on the other hand, consumers' privacy attitudes and behaviors do not always align, revealing the phenomenon of the "privacy paradox". Guo Huibin et al. (2006) [18] found that in a monopolistic competitive market, a good reputation can reduce consumers' "perceived risk" (uncertainty caused by the invisibility of services) towards products, thereby enhancing consumers' purchasing intention. Gong Shiyang et al. (2018) [19] found products with a high-star rating (4.5 stars or above) have a sales volume that is over 30% higher than those with a low-star rating (3 stars or below). Liu Yang et al. (2024) [20] hold that open interaction (the timely response and transparent communication of enterprises to customer feedback) plays a partial mediating role between online word-of-mouth and enterprise performance - word-of-mouth promotes in-depth interaction between enterprises and customers, and is transformed into higher customer loyalty and repeat purchase rates. Park Young Eun and Son Hyunsang (2025) [21] investigated the effects of recommender type and consumption goal (utilitarian versus hedonic) on consumer responses to travel destination recommendation email advertisements.

## 2.2  The effect of information channels on consumer-generated advertising

For companies, information channels significantly affect consumers' willingness to generate advertisements. Existing research indicates that the social attributes and trust foundation of information channels are core drivers of consumer-generated advertising. Yao Xi and Jian Yufan (2016)[66] argued that social drivers directly affect the behavior of consumer-generated content. According to the trust transmission mechanism, consumers develop trust in products based on endorsements and recommendations from others, thereby affecting their decisions and behaviors. When faced with different information channels, consumers weigh perceived benefits against social influence. Zeng Fue et al. (2022) [22] investigated the effectiveness of agentic and communal advertising appeals delivered through two communication channels (private messages and public feeds) on social media. Agentic appeals delivered through public feeds generate favorable advertisement attitudes, communal appeals delivered through private messages engender positive advertisement attitudes.

Recommendations from friends and family, as a form of social effect, are typically regarded as a more credible and reliable source of information, significantly boosting consumers' willingness to generate advertisements. For example, WeChat Moments' "Nearby Ads" feature precisely matches location data with social interactions, making the ads content more readily accepted and shared. When consumers receive information through personal recommendations, they are more inclined to trust and accept it, thereby increasing their willingness to generate advertisements. Therefore, this paper proposes Hypothesis H1.

**H1:** Information channels significantly affect consumer-generated advertising.

## 2.3 The effect of purchase experience on consumer-generated advertising

For companies, distinguishing consumers with different purchasing experiences can be highly beneficial in promoting consumer-generated advertising. Brakus (2009) et al [23] argue that actual consumption experiences, especially highly interactive ones, can strengthen users' brand identification through emotional resonance and memory. Liu Yang et al. (2024) [24] hold that open interaction (the timely response and transparent communication of enterprises to customer feedback) can be transformed into higher customer loyalty and repeat purchase rates. For example, a consumer's purchase experience with a particular branded may create a favorable impression of that brand, thereby increasing the consumer's trust, and consequently, the likelihood of the consumer's participation in that brand activity. Therefore, this paper proposes Hypothesis H2.

**H2:** Purchase experiences significantly affect consumer-generated advertising.

Empirical analysis of Suning.com reveals that consumer satisfaction with purchased products, based on consumer review data, exhibits a correlation coefficient of 0.52($p < 0.1$) with advertising engagement. This indicates that consumers' purchase experiences significantly affect their willingness to purchase products based on the information they obtain. However, Zhu Qiang et al. (2018) [25] found that when consumers acquire product information through recommendations from friends and family, their self-efficacy enhances its moderating effect on risk perception, further weakening the direct effect of purchase experiences. Simultaneously, when evaluating brand credibility, consumers may downplay their own direct purchase experience and instead rely on others' experiences and recommendations. This trust transmission mechanism allows brands not personally purchased to achieve credibility levels approaching those of purchased brands through others' endorsement and recommendation. Therefore, this paper proposes Hypothesis H4.

**H4:** Information channels affect the role of purchase experiences on consumer-generated advertising.

## 2.4 The effect of privacy costs on consumer-generated advertising

The decisions to generate consumer-generate advertising is fundamentally a dynamic trade-off between privacy costs and anticipated benefits. According to privacy computation theory, consumers decide whether to participate in ad-generating activities by weighing "perceived benefits" (such as financial rewards, social capital accumulation) against "perceived risks" (privacy leakage and loss of social evaluation) [25]. Users' perceptions of privacy and security safeguards while using of media technologies will affect their willingness to generate content. The effect of reward level on consumer generated advertising is moderated by privacy costs [3]. When advertising involves sensitive information (such as geographic location, biometric data), consumers may overestimate the potential loss from privacy breaches (such as identity theft or damaged social reputations), forming a loss-dominated decision-making framework. In such cases, even increased rewards may fail to overcome consumers' risk-averse tendency, thereby suppressing their participation. Therefore, this paper proposes Hypothesis H3.

**H3:** Privacy costs significantly affect consumer-generated advertising.

According to the theory of face preservation, individuals strive to maintain their own image in social interactions, avoiding negative evaluations stemming from inappropriate behaviors [26]. Particularly in "strong relationship" contexts, users are more concerned that highly privacy-invasive advertisements such as image-text posts containing purchase records may excessively expose personal details, thereby triggering negative judgments from friends and family regarding their "self-serving sharing". Studies indicates that perceived privacy risk significantly affects privacy-protective behaviors. When users perceive potential privacy breaches, they proactively adopt measure to protect their privacy, such as reducing sharing behaviors on social media [27]. Since consumers are more sensitive to how friends and family perceive their actions, such privacy concerns are particularly pronounced within "strong-tie" networks, therefore, this paper proposes Hypothesis H5.

**H5:** Information channels affect the roles of privacy costs on consumer-generated advertising.

For customers who have never purchased a particular brand, the primary factor affecting consumers-generated advertising is the privacy cost. Compared with high privacy costs, low privacy costs are more effective at motivating consumers

to generate advertisements. For consumers who have previously purchased a brand, if their purchase experience was high positive, this fosters trust in the brand, thereby reducing concerns about privacy risk. In such case consumers primarily focus on the benefits they receive (such as rewards or discounts). Conversely, if the purchase experience was negative, consumers are generally reluctant to generate advertisements. Even if they are willing to do so, their motivation primarily stem from the rewards offered. Therefore, this paper proposes Hypothesis H6.

**H6:** Purchase experiences affect the role of privacy costs on consumer-generated advertising.

### 2.5 The effect of purchase experiences on the relationship between information channels and consumer-generated advertising

For consumers who have previously purchased products, information channels exert a weaker moderating effect on the privacy costs. This is because consumers' historical purchasing behavior reflects their acceptance of the privacy costs associated with such actions, meaning they have already accepted them, or they have lower sensitivity to advertising. Consequently, their willingness to generate advertisements is only minimally affected. For customers who have never purchased the product, information channels exert a stronger moderating effect on privacy cost. This is because non-purchasing consumers derive their privacy concerns entirely from information channels, which significantly affects their perception of the privacy costs. Furthermore, non-purchasing consumers may be more sensitive to privacy costs, leading to a greater effect on their willingness to generate advertisements. Therefore, this paper proposes Hypothesis H7.

**H7:** Consumers' purchase experiences affect the moderating effect of the information channels on privacy costs. Specifically, for customers who have previously purchased products, information channels can not moderate the effect of privacy costs on consumer-generated advertising; for customers who have never purchased products, information channels can moderate the effect of privacy costs on consumer-generated advertising.

The research framework of the paper is shown in Fig 1.

Using an experimental approach to collect data, we test the effects of information channels, purchase experiences and privacy costs on consumer-generated advertising (H1, H2 and H3). Concurrently, we examine the effect of information channels on the relationship between purchase experiences and consumer-generated advertising (H4), as well as the effect of information channels and purchase experiences on the relationship between privacy costs and consumer-generated advertising (H5 and H6).

## 3 Experimental design

When ANOVA was used to analyze consumer recommendations and word-of-mouth marketing, it can help identify the effect of different factors on purchase intention or sales volume, which provide a basis for enterprises to optimize their strategies. When comparing the effects of different marketing strategies or the word-of-mouth effects of different product

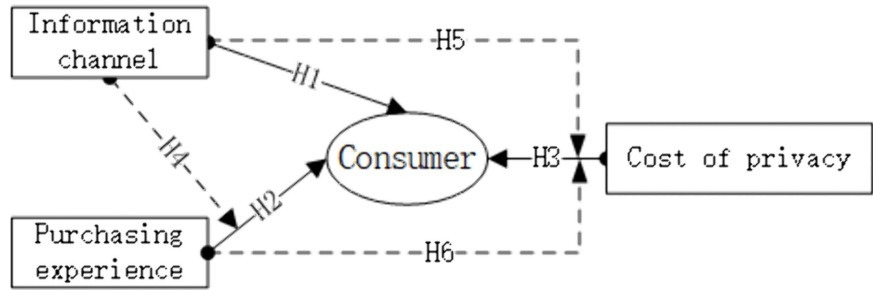

**Fig 1. The research framework of this paper.**

categories, ANOVA can handle multiple groups simultaneously, avoiding the accumulation of errors caused by multiple t-tests, which is very common in consumer recommendation and word-of-mouth marketing research. Therefore, we designed an ANOVA variance analysis experiment to handle the effect of multiple factors by comparing the mean differences of multiple sets of data, and to analyze the interaction between two independent variables and the dependent variable.

### 3.1 Experiment 1: The effect of purchase experiences and information channels on consumer-generated advertising

**3.1.1 Experimental design.** Experiment 1 used between-groups factorial design of 2 (purchase experiences: Purchased vs. Not purchased) × 2 (information channels: Self-discovered vs. Recommended from friends and family). We collected 402 valid questionnaires through a company's online questionnaire system, in which, 204 were male, accounting for 50.75%; 198 were female, accounting for 49.25%, and $M_{age} = 24.47$. There is no minor in participants. And the experiment was conducted using a mini-program (QuestionStar in Wechat) to carry out questionnaires on electronic devices. The questionnaire results cannot be collected if the participants do not consent. Nanjing Forest university approved this study. And the form of consent is obtained by written.

**3.1.2 Pre-experimental measurement.** To control the potential effect of the difference between offline and online platform on the results, we conducted a test on "online or offline platforms". The independent samples t-test showed there was no significant difference in recommendation willingness between the two types of platforms ($M_{offline} = 3.259$, $SD_{offline} = 1.051$, $M_{online} = 3.6$, $SD_{online} = 1.087$, $F = 0.753, p = 0.386$).

**3.1.3 Experimental flow.** After entering the experiment, participants were randomly assigned to four scenarios for questionnaire responses. Scenario 1 is "Self-discovered + Purchased", i.e., participants needed to imagine themselves seeing promotional advertisements of milk tea brands they had previously purchased on online platforms or in physical stores based on their life experiences. They could get discount if they helped recommend the advertisements. Scenario 2 is "Self-discovered + Not purchased", i.e., participants needed to imagine themselves seeing promotional advertisements of milk tea brands they had never purchased on online platforms or in physical stores based on their life experiences. They could get discount if they helped recommend the advertisements. Scenario 3 is "Recommended from friends and family + Purchased", i.e., participants needed to imagine activities where their friends and family told them about the milk tea brand that they had previously purchased based on their life experiences. They could get discount if they helped recommend the advertisement. Scenario 4 is "Recommended from friends and relatives + Not purchased", i.e., participants needed to imagine activities where their friends and family told them about the milk tea brand that they had never purchased based on their life experiences. They could get discount if they helped recommend the advertisement.

The experiment began with a manipulation test. In the dimension of consumer purchase experiences, it was concluded that the score of perceived value of purchased group was significantly higher than that of not purchased group ($M_{Had\ Purchased} = 3.152 < M_{Had\ Not\ Purchased} = 3.483$, $T = 2.654$, $P = 0.008***$), therefore the manipulation of the distinction of purchase experiences in this experiment was successful, i.e., the hypotheses about purchase experiences in this experiment were reasonable; in the dimension of information channels, it was concluded that: The score of perceived value of friends and family recommendation group was significantly higher than that of Self-discovered group ($M_{autonomous} = 3.007 < M_{friends\ and\ family} = 3.415$, $T = -2.922$, $P = 0.004***$), so the differentiation manipulation of information channels in this experiment was successful, i.e., the hypotheses about information channels in this experiment were reasonable.

**3.1.4 Experimental results.** An analysis of variance (ANOVA) of 2 (purchasing experiences: Purchased vs. Not purchased)× 2 (information channels: Self-discovered vs. Recommended from friends and family) was performed on the participants, and the results are shown in Table 1.

**Table 1. Effect of purchase experiences and information channels on consumer-generated advertising.**

| Terms | Square Sum | Degrees of Freedom | Mean Square | F | P | R² | Adjustment of R² |
|---|---|---|---|---|---|---|---|
| Intercept (The Point at Which a Line Crosses The x- or y-Axis) | 6806.462 | 1 | 6806.462 | 1882.87 | 0.000*** | 0.05 | 0.043 |
| Information Channels | 36.469 | 1 | 36.469 | 10.089 | 0.002*** | | |
| Purchase Experiences | 18.566 | 1 | 18.566 | 5.136 | 0.024** | | |
| Information Channels * Purchase Experiences | 52.083 | 1 | 52.083 | 14.408 | 0.000*** | | |

**3.1.5 Experimental analysis.** According to Table 1, we conclude that **the main effect of information channels is significant** (F = 10.089, P = 0.002***), i.e., compared with consumers seeking information independently, recommendation from friends and family are more effective at motivating consumers to generate advertisements, thereby validating Hypothesis H1. Simultaneously, according to Table 1, we also conclude that **the main effect of purchase experiences is significant** (F = 5.136, P = 0.024**), i.e., compared with the unpurchased brands, the purchased brands are effective at motivating consumers to generate advertisements, thereby validating Hypothesis H2. Additionally, according to Table 1, we can also conclude that **purchase experiences interact with information channels**, i.e., information channels can modulate the effect of purchase experiences on consumer-generated advertising.

To further analyze the moderating effect of information channels, we conducted a simple effects analysis to examine the data differences of consumer-generated advertising in different information channels and different purchase experiences. The results are shown in Fig 2: **First, the main effect of purchase experience is significant** (F = 15.412, P = 0.000***) **when consumers independently obtain information.** For example, if consumers have prior positive purchase experiences, they will have a deeper understanding and trust in this brand, which will reduce the perceived risk in purchasing decision. This, in turn, encourages more consumers to participate in activities and to increase product purchase. When consumers have never purchased the brand's products, they lack direct experience with the brand's offerings. This means they cannot evaluate the brand's quality, performance or service based on personal purchase

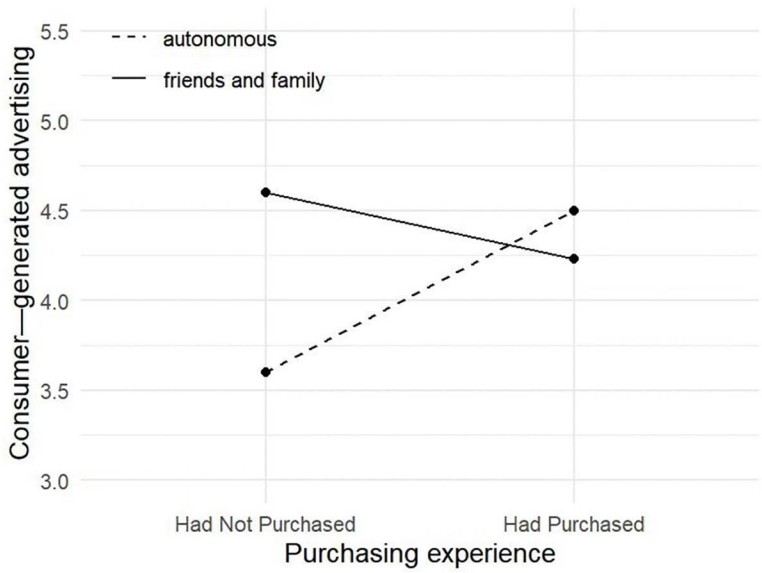

**Fig 2. Interaction between purchase experiences and information channels.**

experiences, potentially leading to higher perceived risk for the brand's products. This risk perception may affect their purchase decisions, making them more cautious or hesitant. **Second, the main effect of consumers' purchase experiences is insignificant when customers make purchases base on recommendation from friends and family** (F = 4.925, p = 0.027**). The reason may be that when consumers get advertisements through the recommendation from friends and family, the source of brand trust shifts from personal experience (purchase experiences) to relational trust (friends and family reliability). For example, when an unpurchased brand is endorsed by friends and family, consumers may reason that "the brand must be useful to friends and family", thus weakening the importance of their own purchase experience.

Therefore, the effect of purchase experiences on consumer-generated advertising is significant when customers independently obtain information. However, the effect of purchase experiences on consumer-generated advertising is not significant when purchases are made based on recommendations from friends and family, which validating Hypothesis H4.

**3.1.6 Conclusions of the study.** Based on the above research, this paper draws the following conclusions:

**Conclusion 1**: Compared with products they purchased themselves, consumers are more willing to generate ads for products recommended from friends and family; customers with purchase experiences for a brand are more willing to generate ads than those without such experience.

**Conclusion 2**: Information channels affect the relationship between purchase experiences and consumer-generated advertising. For products consumers acquired themselves, their purchase experiences significantly affects their willingness to generate advertisements; however, for products recommended from consumers' friends and family, their purchase experiences does no significantly affect their willingness to generate advertisements.

## 3.2 Experiment 2: The effect of information channels and privacy costs on consumer-generated advertising

**3.2.1 Experimental design and pre-experimental measurement..** Experiment 2 used between-groups factorial experimental design of 2 (information channels: Self-discovered vs. Recommended from friends and family) × 2 (privacy costs: high vs. low). The participants of the experiment were the same as in Experiment 1.

**3.2.2 Experimental flow..** After entering the experiment, participants were randomly assigned one of four scenarios for questionnaire responses. Scenario 1 was "Self-discovered + High privacy costs", i.e., participants imagined seeing an advertisement of milk tea brand on an online platform or in a physical store based on their life experiences. They could receive a discount if they helped recommend the advertisement, and the form of recommendation was "Sharing the advertisement with image and text to their WeChat Moments"; Scenario 2 was "Recommended from friends and family + High privacy costs", i.e., participants imagined that their friends and family informed them about promotions from a milk tea brand they had previously purchased. They could receive a discount if they helped recommend the advertisement, and the form of recommendation was "Sharing the advertisement with images and text on their WeChat Moments"; Scenario 3 was "Self-discovered + Low privacy costs", i.e., participants imagined encountering a milk tea brand's promotional advertisement on an online platform or in a physical store based on their life experiences. They could receive a discount if they helped recommend the advertisement, and the form of recommendation was "Sharing the advertisement as a text review on WeChat Moments"; Scenario 4 was "Recommended from friends and family + Low privacy costs", i.e., participants imagined that their friends and family inform them about promotions from a milk tea brand they had previously purchased. They could receive a discount if they helped recommend the advertisement, and the form of recommendation was "Sharing the advertisement as a text review on WeChat Moments".

First, we conduct a manipulation test on privacy cost through an independent samples t-test. and the study shown compared with text reviews, image-and-text reviews enable participants to perceive more privacy costs ($M_{high\ cost}$ = 3.757, $SD_{high\ cost}$ = 2.007, $M_{low\ cost}$ = 4.614, $SD_{low\ cost}$ = 1.714, T = −4.034, p = 0.000* **). Therefore, the manipulation of privacy cost levels in this experiment was successful, validating the hypothesis regarding privacy cost in this experiment.

### 3.2.3 Experimental results.

An analysis of variance (ANOVA) of 2 (information channels: self-discovered vs. Recommended from friends and family) × 2 (privacy costs: high vs. low) was performed on the participants, and the results are shown in Table 2.

### 3.2.4 Experimental analysis.

According to Table 2, we conclude that **the main effect of privacy costs is significant** (F = 5, P = 0.026**), i.e., privacy costs significantly affects consumer-generated advertising. Compared with high privacy costs, low privacy costs more effectively motivate consumers to generated advertisements, validating Hypothesis H3. Additionally, Table 1 indicates that **the main effect of information channels is significant** (F = 4.693, P = 0.031**), i.e., recommendations from friends and family motivate consumers to generate advertisements more effectively than self-initiated information acquisition.

Additionally, according to Table 2, we can also find **a significant interaction between information channels and privacy costs** (F = 8.14, p = 0.005***), i.e., information channels modulate the effect of privacy costs on consumer-generated advertising.

Then, this paper conducts a simple effects analysis to examine data differences of consumer-generated advertising in different product types and reward levels. The results are shown in Fig 3: **First, in the context of recommendation from friends and family, the privacy costs significantly inhibits willingness to recommend** (F = 9.842, p = 0.002***), i.e., in

**Table 2. Effect of information channels and privacy costs on consumer-generated advertising.**

| Terms | Equation of Squares | Degrees of Freedom | Mean Square | F | P | R² | Adjustment of R² |
|---|---|---|---|---|---|---|---|
| **Intercept (The Point at Which a Line Crosses The x- or y-Axis)** | 6346.05 | 1 | 6346.05 | 1613.649 | 0.000*** | 0.033 | 0.026 |
| **Information Channels** | 18.455 | 1 | 18.455 | 4.693 | 0.031** | | |
| **Privacy Costs** | 19.663 | 1 | 19.663 | 5 | 0.026** | | |
| **Information Channels * Privacy Costs** | 32.014 | 1 | 32.014 | 8.14 | 0.005*** | | |

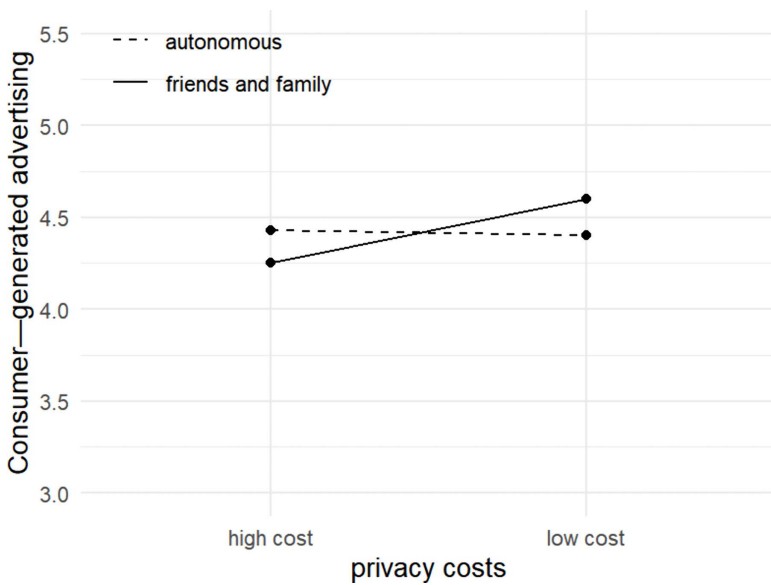

**Fig 3. Interaction between information channels and privacy costs.**

the context of recommendation from friends and family, compared with low privacy costs, high privacy costs inhibit consumers' willingness to generate advertisements. This may stem from face concerns in "strong relationship" contexts, i.e., users worry that advertisements with high privacy costs (e.g., graphics containing purchase records) may overly expose personal details, potentially inviting negative evaluations from friends and family regarding their "utilitarian sharing". **Second, the main effect of privacy costs becomes insignificant in the context of self-discover** (F = 0.261, p = 0.610), i.e., when consumers actively seek information, high privacy costs actually motivate them to generate advertisements more than low privacy costs do. This may occur because consumers perceive sharing advertisements with high privacy costs as a form of identity signals (e.g., showcasing their tastes in consumption, flaunting wealth, or expressing their individuality), and this need for self-presentation overrides privacy concerns.

Therefore, privacy costs significantly affect consumer-generated advertising when consumer was recommended from friends and family. However, the privacy cost has no significant effect on consumer-generated advertising when consumers proactively seek out the content, validating Hypothesis H5.

**3.2.5 Conclusions of the study..** Based on the above research, this paper presents the following conclusions:

**Conclusion 3**: Information channels affect the relationship between privacy costs and consumer-generated advertising. Consumers are more willing to generate advertisements for product with low privacy costs when the product information was recommended from friends and family; and consumers more willing to generate advertisements for product with high privacy costs when the product information was obtained independently by consumers.

### 3.3 Experiment 3: The effect of purchase experiences and privacy costs on consumer-generated advertising

**3.3.1 Experimental design..** Experiment 3 use between-groups factorial experimental design of 2 ((purchase experiences: ever vs. never) x 2 (privacy: high vs. low). The participants were the same as in Experiment 1.

**3.3.2 Experimental flow..** After entering the experiment, participants could randomly select one of four scenarios for questionnaire responses. Scenario 1 is "Purchased + High privacy costs", i.e., participants needed to imagine, based on personal experiences, seeing promotional advertisement of a milk tea brand they had purchased on an online platform or in a physical store. They could receive a discount if they helped recommend the advertisement, and the form of recommendation was "Sharing the advertisement with image and text to their WeChat Moment". Scenario 2 was "Unpurchased + High privacy costs", i.e., participants needed to imagine encountering promotional advertisement of a milk tea brand they had never purchased, either online or in physical store. They could receive a discount if they helped recommend the advertisement, and the form of recommendation was "Sharing the image-test advertisement to the their WeChat Moments". Scenario 3 was "Purchased + Low privacy costs", i.e., participants imagined seeing promotional advertisement of a milk tea brand they had previously purchased on online platforms or in physical stores. They could receive a discount if they helped recommend the advertisement, and the form of recommendation was "Sharing the advertisement as a text review on WeChat Moments". Scenario 4 was "Unpurchased + Low privacy costs", i.e., participants imagined encountering promotional advertisement of a milk tea brand they had never purchased, either online or in physical stores. They could receive a discount if they helped recommend the advertisement, and the form of recommendation was "Sharing the advertisement as a text review on WeChat Moments".

**3.3.3 Experimental results..** From Experiments 1 and 2 we can conclude that the settings of privacy costs and purchase experiences are reasonable.

An analysis of variance (ANOVA) of 2 (purchase experiences: ever vs. never)× 2 (privacy costs: high vs. low) was performed on participants and the results are shown in Table 3.

**3.3.4 Experimental analysis..** According to Table 3, we conclude that **purchase experiences significantly affect consumer-generated advertising** (F = 18.88, P = 0.000***), i.e., purchased brands motivate consumers to generated advertisements more effectively than unpurchased brands, thereby validating Hypothesis H2. Additionally Table 3

**Table 3. Effect of purchase experiences and privacy costs on consumer-generated advertising.**

| Terms | Equation of Squares | Degrees of Freedom | Mean Square | F | P | R² | Adjustment of R² |
|---|---|---|---|---|---|---|---|
| Intercept (The Point at Which a Line Crosses The x- or y-Axis) | 5783.075 | 1 | 5783.075 | 1823.069 | 0.000*** | 0.07 | 0.063 |
| Purchase Experiences | 59.889 | 1 | 59.889 | 18.88 | 0.000*** | | |
| Privacy Costs | 33.52 | 1 | 33.52 | 10.567 | 0.001*** | | |
| Purchase Experiences* Privacy Cost | 14.97 | 1 | 14.97 | 4.719 | 0.030** | | |
| Purchase Experiences * Information Channels * Privacy Costs | 2.105 | 1 | 2.105 | 0.878 | 0.350 | | |

indicates that **privacy costs significantly affects consumer-generated advertising** (F = 10.567, P = 0.001***), i.e., low privacy costs motivate consumers to generate advertisements more effectively than high privacy costs, thereby validating Hypothesis H3.

In addition, as shown in Table 1, we also conclude that **purchase experiences interact with privacy costs** (F = 4.719, p = 0.030**), i.e., purchase experiences moderates the effect of privacy costs on consumer-generated advertising.

To further analyze the moderating effect of information channels, we conducted a simple effects analysis to examine data differences of consumer-generated advertising in different information channels and purchase experiences. The results are shown in Fig 4: **First, the main effect of privacy costs is significant when consumers have never purchased a brand's products** (F = 8.654, p = 0.004***). This may stem from consumers lacking prior experience with new brands, leading to information asymmetry regarding the brand's privacy protection capabilities. In such cases, privacy risk are amplified in consumers' mind, becoming a primary consideration. For example, when users first register on a social platform, their sensitivity toward authorizing personal privacy (e.g., cell phone contacts, photos, etc.) is higher than on platforms they have used long-term. **Second, the main effect of privacy costs is not significant when consumers have previously purchased a brand's products** (F = 1.098, p = 0.296). Since negative purchase experiences rarely

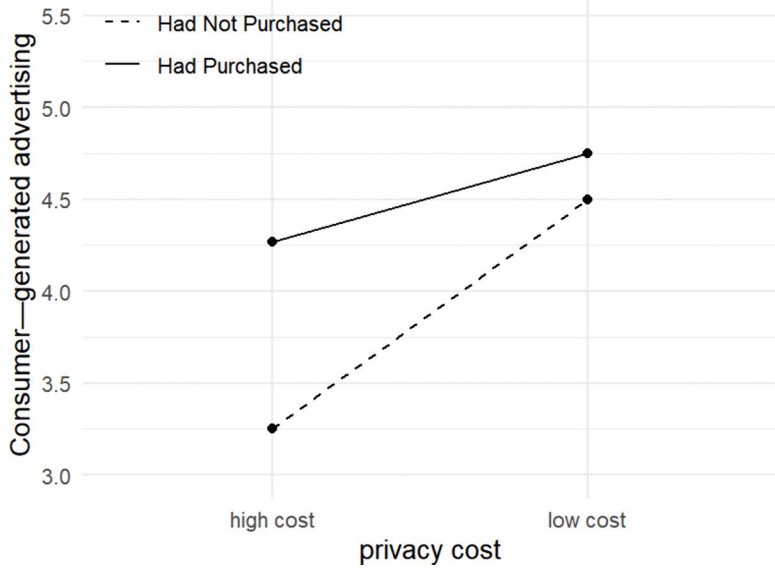

**Fig 4. Interaction between purchase experience and privacy cost.**

prompt consumers to generate advertisements, only scenarios with positive purchasing experiences are analyzed. A positive purchase experience fosters consumer trust in the brand. This trust reduces consumers' concerns about privacy risks, shifting their focus toward the benefits they receive (e.g., rewards or discount, etc.). For example, users who have long relied on a platform are more willing to consent when the app requests additional biometric information for enhancing convenience, prioritizing convenience and trust over privacy costs.

Therefore, for consumers who have never purchased, privacy costs significantly affect consumer-generated advertising; for consumers who have previously purchased, the privacy costs does not significantly affect consumer-generated advertising, validating Hypothesis H6.

**3.3.5 Conclusions of the study..** Based on the above research, this paper presents the following conclusions:

**Conclusion 4**: Purchase experiences affect the relationship between privacy costs and consumer-generated advertising. When consumers have never purchased a brand, privacy costs significantly affect their willingness to generate advertisements; when consumers have previously purchased a brand, privacy costs do not significantly affect their willingness to generate advertisements.

## 3.4 Experiment 4: Interaction of purchase experiences, information channels and the privacy costs

From Table 2, the interaction between purchase experiences, information channels and privacy costs is not significant ($F = 0.878$, $p = 0.350$). This indicates that consumers' purchasing experiences do not affect the moderating effect of information channels on privacy costs. Therefore, Hypothesis H7 is rejected. Consequently, this paper concludes that the effectiveness of consumer-generated advertising is jointly affected by multiple factors, but the interaction between purchase experiences, information channels and privacy costs is not significant. The strong dominance of the information channels (e.g., the credibility of referral channels from acquaintances) and the widespread prevalence of privacy sensitivity may obscure differences arising from purchase experiences. Simultaneously, consumers' rational filtering within complex information environment weakens the moderating effect of purchase experiences. Furthermore, this may be because the inherent characteristics of advertisements (e.g., content, presentation, etc.) playing a more critical role in affecting consumers' responses, while factors like consumers' purchase experiences, information channels, and privacy costs exert relatively weaker effect or contract each other. Based on this, the paper presents the following conclusions:

**Conclusion 5:** Among the various factors affecting consumers' willingness to generate advertisements, although purchase experiences, information channels, and privacy costs exhibit significant interactions with each other, the interaction between these three factors was not significant.

## 4 Conclusions and recommendations

Although the factors affecting consumer-generated advertising have been widely studied, current research primarily adopts a corporate perspective, overlooking considerations of consumers' willingness and underlying psychological mechanisms. From the consumers' perspective, consumer-generated advertising preferred over other forms of advertising for its perceived authenticity and reliability, although this comes at the cost of consumer privacy. Additionally, the consumers' purchasing experiences and the channels through which they obtain product information affect their trust in products, which in turn affects their willingness to generate advertisements.

This study employs questionnaires and experiments simulating real-world consumption scenarios to examine how purchase experiences (Purchased vs. Not purchased), information channels (Self-discovered vs. Recommended from friends and family), and privacy costs (Low vs. High cost) affect consumer-generated advertising. **Theoretically**, it establishes a new research perspectives and domain within consumer-generated advertising, enriching and refining the theoretical framework; **Practically,** it offers insights for enterprises managing consumer-generated advertising, providing marketing guidance on whether or not to utilize it and how to effectively manage it.

**4.1 Conclusions of the study**

This paper proposes and tests research hypotheses in five aspects. Results indicated that information channels, purchase experiences, and privacy costs significantly affect consumer-generated advertising. The effect of purchase experiences on consumer-generated advertising is moderated by information channels, while the effect of privacy costs on consumer-generated advertising is moderated by information channels and purchase experiences. Specifically:

(1) Compared with products they acquired themselves, consumers are more willing to generate advertisements for products recommended from friends and family. Consumers with purchase experiences for a brand are more willing to generate advertisements than those without purchase experiences. Low privacy costs motivate consumers to generate advertisements more effectively than high privacy costs.

(2) Information channels affect the relationship between purchase experiences and consumer-generated advertising. For products consumer acquired through their own purchase experiences, these purchase experiences significantly affect their willingness to generate advertisements, consistent with the findings in (1). However, for products recommended from friends and family, the effect of consumers' purchase experiences on their willingness to generate advertisements is insignificant, i.e., personal purchase experiences lose importance when recommendations come from friends and family. This effect likely arises because when consumers encounter advertisements through the channel of friends and family, the source of brand trust shifts from personal experiences (purchase experiences) to relational trust (friends and family reliability).

(3) Information channels affect the relationship between privacy costs and consumer-generated advertising. For products recommended from friends and family, consumers are more willing to generate advertisements for products with low privacy costs, consistent with the findings in (1). However, for products consumers acquired through their own purchase experiences they are more willing to generate advertisements for products with high privacy costs. This may stem from consumers viewing the forwarding of high-privacy-costs advertisements as identity signals (e.g., showcasing tastes or displaying wealth, or reflecting personalization), and this self-presentation need overrides privacy concerns.

(4) Purchase experiences affect the relationship between privacy costs and consumer-generated advertising. If consumers have never purchased a brand, privacy costs significantly affect their willingness to generate advertisements, consistent with the findings in (1). However, if consumers have previously purchased a brand, privacy costs do not significantly affect their willingness to generate advertisements. This may occur because positive purchase experiences foster consumer trust in the brand. Such trust reduces consumers' concerns about privacy risks, shifting their focus toward the benefits they receive (e.g., rewards, discounts, etc.).

(5) Among the various factors affecting consumers' willingness to generate advertisements, while purchase experiences, information channels, and privacy costs exhibit significant interactions with each other, the interaction among the three factors is not significant.

**4.2 Management**

(1) When implementing consumer-generated advertising, enterprises should increase investment in referral programs through friends and family while enhancing rewards for advertisements generated by existing customers. For example, enterprises can leverage social networks to amplify advertising reach through referral strategies. Additionally, they can strengthen social sharing mechanism by designing incentive programs to encourage existing users to share advertising content.

(2) If an enterprise's consumer base primarily obtains products information through recommendations from friends and family, the enterprise need not adopt differentiated incentive strategy for generating advertisements targeting new

and existing consumers. If an enterprise's consumer base primarily obtains products information through their own proactive efforts, it is recommended that the enterprise adopt differentiated incentive strategies for generating advertisements targeting new and existing customer. Compared with existing customers, enterprise should increase the investment in optimizing the brand trust strategy for new customers. For example, establish initial brand trust among non-purchasing users through the recommendation from friends and family, thereby reducing the effect of the purchase experiences on brand perception, and converting non-purchasing users into potential consumers.

(3) When consumers primarily obtain product information through recommendations from friends and family, enterprises should prioritize consumer privacy protection in their consumer-generated advertising strategy, i.e., adopting a low privacy cost strategy. Conversely, when consumers mainly obtain product information independently, enterprises may appropriately leverage consumer privacy for advertising purposes, i.e., high privacy costs strategy.

For example, in the scenarios of recommendation from friends and family, users prioritize privacy protection. Anonymizing recommended content can effectively alleviate privacy concerns, thereby boosting sharing willingness. Conversely, in the scenarios of self-initiated dissemination, users tend to showcase personalized content. Recommendation systems should leverage this display demand by providing more tailored and engaging content to encourage voluntary sharing behavior.

(4) Given inherent tension between personalized services and privacy protection, enterprises should adopt the following strategies: For existing customers, fully collect and leverage consumers' privacy data while increasing investment in personalized services to enhance positive experiences and encourage consumers voluntarily generate advertisements. For new customers, minimize the collection of sensitive information, while strengthening privacy protection to alleviate consumers' privacy concerns.

For example, when targeting new customers, enterprises can enhance customers trust by clearly informing them of the purpose, scope, and protective measures for collecting personal information through transparent privacy policies and user agreements. For existing customers, enterprises can gather additional privacy information by hosting online and offline interactive activities, such as lucky draws, Q&A sessions, and live broadcasts. Leveraging social media platforms to deliver more personalized services boosts customers satisfaction, thereby stimulating consumers engagement and generating desire for advertising.

(5) When formulating incentive strategies for consumer-generated advertising, enterprises need only consider the interactions between any two of the following factors: referral rewards, privacy costs, purchase experiences, and information channels. There is no need to account for interactions among all three factors.

### 4.3 Research outlook

Although the management insights presented in this paper hold practical significance, certain limitations exist. First, the participants in this paper were relatively homogeneous, with a limited sample size predominantly consisting of young adults, which may affect the generalizability and applicability of the findings. Secondly, many experimental scenarios could not replicate real-world conditions, relying primarily on participants' imagination, which could introduce some errors. Future research could validate these conclusions by collecting more representative samples, thereby enhancing the robustness of the findings. Third, the non-significant three-way interaction is interesting, which will be the future research.

### Supporting information

**S1 Table. Purchase experience and information channels.**
(XLSX)

**S2 Table. Information channels and privacy cost.**
(XLSX)

**S3 Table. Privacy cost and purchase experience.**
(XLSX)

## Author contributions

**Data curation:** Yu LIU.

**Formal analysis:** Chuanxi Cai.

**Funding acquisition:** Chuanxi Cai.

**Investigation:** Yu LIU.

**Methodology:** Yu LIU.

**Resources:** Chuanxi Cai.

**Supervision:** Chuanxi Cai.

**Validation:** Chuanxi Cai.

**Writing – original draft:** Yu LIU.

**Writing – review & editing:** Chuanxi Cai.

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
