## [Decision Letter · Decision Letter 0]

20 Oct 2025

Dear Dr. Cai,

Thank you for submitting your manuscript to PLOS ONE. After careful consideration, we feel that it has merit but does not fully meet PLOS ONE’s publication criteria as it currently stands. Therefore, we invite you to submit a revised version of the manuscript that addresses the points raised during the review process.

We look forward to receiving your revised manuscript.

Kind regards,

Reza Rostamzadeh

Academic Editor

PLOS ONE

Journal Requirements:

2. Please provide details regarding participant consent. In the ethics statement in the Methods and online submission information, please ensure that you have specified (1) whether consent was informed and (2) what type you obtained (for instance, written or verbal, and if verbal, how it was documented and witnessed). If your study included minors, state whether you obtained consent from parents or guardians. If the need for consent was waived by the ethics committee, please include this information.

This work was supported in part by The Science and Technology Innovation Fund (163060171) and in part by the General Program in philosophy and Social Sciences (2022SJYB0124)

5. We note that your Data Availability Statement is currently as follows: All relevant data are within the manuscript and its Supporting Information files.

Additional Editor Comments:

In consideration of the references, the authors mainly depend on Chinese database of journals and research. While there are many researches have been written in this topic, especially consumers’ experience and recommendations to which can be benefited by companies in different methods of promotion.

Reviewers' comments:

Reviewer's Responses to Questions

**Comments to the Author**

1. Is the manuscript technically sound, and do the data support the conclusions?

Reviewer #1: Partly

Reviewer #2: Partly

2. Has the statistical analysis been performed appropriately and rigorously?

Reviewer #1: Yes

Reviewer #2: I Don't Know

3. Have the authors made all data underlying the findings in their manuscript fully available?

Reviewer #1: No

Reviewer #2: No

4. Is the manuscript presented in an intelligible fashion and written in standard English?

Reviewer #1: Yes

Reviewer #2: Yes

Reviewer #1: Abstract:

- As a beginning, the abstract should be revised to focus on the most important information that determine research process and brief of the results. Also, give a brief of research methodology and sample size for each experiment. Make it short for the reader.

Introduction:

- There are several statements that have not been referenced. Those either to be cited and should be referenced or by the authors and must be justified. For example, “At the same time a lot of more people tend to trust the information provided by consumers rather than the information provided by companies”? At any level of introduction, theoretical, and the literature review whenever that is statement as a fact should be referenced or justified.

Theoretical Analysis and Research Hypothesis:

- When proposing a hypothesis, there is no need for more information and explanation. This further information may interrupt focus on testing hypothesis. First statement of each hypothesis is enough.

Experimental Design:

- Normally, there should be a separate section that determines research methodology and research design and then a sperate section for analysis part. This type of writing gives a clear vision of how data will be analysed.

- I am not quite happy using ANOVA to test hypothesis. This kind of relationship should be examined using PLS technique to examine whole model, especially when there is a mediating factor that must be examined in designated model. However, it can be accepted in this case due to

Conclusions and Recommendations:

- “This paper proposes and tested five research hypotheses”. There were 6 hypotheses proposed and tested. Even insignificant results should be presented.

References:

- As far as I know, there are plenty of research are conducted discovering the relationship between consumer’s recommendations or consumer’s world of mouth through social media and companies that benefit from these recommendations. Number of cited papers is not enough.

Reviewer #2: Theoretical Contribution and Discussion:

The discussion of the non-significant three-way interaction is speculative. While your explanations are interesting, they should be framed more clearly as potential interpretations and avenues for future research rather than definitive conclusions, as they were not measured or tested.

The literature review is adequate but could be more tightly focused on framing the specific gaps your three key variables address. The link from general UGC/CGA motivations to your specific experimental variables could be sharper.

Language and Clarity:

The manuscript must be professionally proofread to correct grammatical errors and improve phrasing. Pay particular attention to verb tenses and article usage.

Minor Comments:

The abstract is very long and detailed. Consider shortening it and focusing on the key objectives, methods, and top-level findings.

The term "commodity" is used throughout. While not incorrect, "product" or "brand" is more common in marketing literature for this context (e.g., milk tea).

Conclusion:

This paper has the potential to be a valuable contribution to the literature on consumer-generated content and advertising. The experimental design is strong, and the research questions are pertinent. However, significant revisions are required to meet the standards of a leading journal. These revisions primarily involve enhancing methodological transparency, providing full data access, improving statistical reporting rigor, and polishing the language.

**Do you want your identity to be public for this peer review?** For information about this choice, including consent withdrawal, please see our Privacy Policy

Reviewer #1: **Yes:** Tawfeeq M. Alanazi

Reviewer #2: No

---

## [Author Response · Author response to Decision Letter 1]

25 Nov 2025

Dear Reviewers:

Thank you for your review, we have amended this article according to your advice. All the question and answer are as follows:

Reviewer #1:

Abstract: - As a beginning, the abstract should be revised to focus on the most important information that determine research process and brief of the results. Also, give a brief of research methodology and sample size for each experiment. Make it short for the reader.

Answer: We have revised the abstract according to your review. For example, we have added the research process, research methodology and sample size for each experiment. Meanwhile, we have streamlined the abstract, retaining only the important information.

Introduction: - There are several statements that have not been referenced. Those either to be cited and should be referenced or by the authors and must be justified. For example, “At the same time a lot of more people tend to trust the information provided by consumers rather than the information provided by companies”? At any level of introduction, theoretical, and the literature review whenever that is statement as a fact should be referenced or justified.

Answer: According to your review, we have added the reference for the relevant statements in introduction, such as following:

[1] Saura Jose Ramon, Dwivedi Yogesh K, and Palacios-Marqués Daniel. Online user behavior and user-generated content[J]. Frontiers in Psychology, 2022, 13: 895467.

[2] Hochstein Rachel E., Harmeling Colleen M., Taylor Perko. Toward a theory of consumer digital trust: Meta-analytic evidence of its role in the effectiveness of user-generated content[J]. Journal of the Academy of Marketing Science, 2025, 53(4): 1034-1054.

Theoretical Analysis and Research Hypothesis: - When proposing a hypothesis, there is no need for more information and explanation. This further information may interrupt focus on testing hypothesis. First statement of each hypothesis is enough.

Answer: We have deleted the explanation of all hypothesis in Theoretical Analysis and Research Hypothesis.

Experimental Design: - Normally, there should be a separate section that determines research methodology and research design and then a separate section for analysis part. This type of writing gives a clear vision of how data will be analysed.

- I am not quite happy using ANOVA to test hypothesis. This kind of relationship should be examined using PLS technique to examine whole model, especially when there is a mediating factor that must be examined in designated model. However, it can be accepted in this case due to..

Answer: According to your review, we have separated the analysis part in the experimental. Additionally, we have added the reasons for adopting ANOVA before the experimental, such as following:

When ANOVA was used to analyze consumer recommendations and word-of-mouth marketing, it can help identify the impact of different factors on purchase intention or sales volume, which provide a basis for enterprises to optimize their strategies. When comparing the effects of different marketing strategies or the word-of-mouth influence of different product categories, ANOVA can handle multiple groups simultaneously, avoiding the accumulation of errors caused by multiple t-tests, which is very common in consumer recommendation and word-of-mouth marketing research(Liu Juan, Zhang Jianqiang and Zhong Weijun 2021; Eelen Jiska et al. 2017; Ullal, Mithun S et al. 2021). Therefore, we designed an ANOVA variance analysis experiment to handle the influence of multiple factors by comparing the mean differences of multiple sets of data, and to analyze the interaction between two independent variables and the dependent variable.

Conclusions and Recommendations: - “This paper proposes and tested five research hypotheses”. There were 6 hypotheses proposed and tested. Even insignificant results should be presented.

Answer: This sentence might be ambiguous. We have made the necessary revisions in manuscript. In fact, what we want to cover is we made the assumptions in five aspects. The first aspect includes H1, H2 and H3. The second aspect is H4, the third aspect is H5, the fourth aspect is H6, and the fifth aspect is H7.

References: - As far as I know, there are plenty of research are conducted discovering the relationship between consumer’s recommendations or consumer’s world of mouth through social media and companies that benefit from these recommendations. Number of cited papers is not enough.

Answer: According to your review, we have added the relevant paper in theoretical analysis and research hypotheses. For example: Guo Huibin et al. (2006) found that in a monopolistic competitive market, a good reputation can reduce consumers' "perceived risk" (uncertainty caused by the invisibility of services) towards products, thereby enhancing consumers' purchasing intention. Gong Shiyang et al (2018) found products with a high-star rating (4.5 stars or above) have a sales volume that is over 30% higher than those with a low-star rating (3 stars or below). Liu Yang et al. (2024) hold that open interaction (the timely response and transparent communication of enterprises to customer feedback) plays a partial mediating role between online word-of-mouth and enterprise performance - word-of-mouth promotes in-depth interaction between enterprises and customers, and is transformed into higher customer loyalty and repeat purchase rates. Park Young Eun and Son Hyunsang (2025) investigated the effects of recommender type and consumption goal (utilitarian versus hedonic) on consumer responses to travel destination recommendation email advertisements.

Reviewer #2:

Theoretical Contribution and Discussion: - The discussion of the non-significant three-way interaction is speculative. While your explanations are interesting, they should be framed more clearly as potential interpretations and avenues for future research rather than definitive conclusions, as they were not measured or tested.

Answer�the non-significant three-way interaction has been put in the future research.

Literature review: - The literature review is adequate but could be more tightly focused on framing the specific gaps your three key variables address. The link from general UGC/CGA motivations to your specific experimental variables could be sharper.

Answer According to your review, we have modified the relevant literature in theoretical analysis and research hypotheses. For example, Zeng Fue et al. (2022) investigated the effectiveness of agentic and communal advertising appeals delivered through two communication channels (private messages and public feeds) on social media. Agentic appeals delivered through public feeds generate favorable advertisement attitudes, Communal appeals delivered through private messages engender positive advertisement attitudes. Liu Yang et al. hold that open interaction (the timely response and transparent communication of enterprises to customer feedback) can be transformed into higher customer loyalty and repeat purchase rates. Liu Juan et al. (2021) found the effect of reward level on consumer generated advertising is moderated by privacy cost. In addition, the less relevant references were also deleted.

Moreover, the existing studies mainly focused on the direct impact and moderating effect of the three key variables (purchase experiences, information channels and privacy costs) on consumer behavior from the perspective of enterprises, without consideration of consumers themselves. However , the motivation for consumer to generate advertisements is mainly determined by their perceived benefits, so we analyze the direct impact and moderating effect of the three key variables on consumer behavior from the perspective of consumers' psychological mechanism.

Language and Clarity: - The manuscript must be professionally proofread to correct grammatical errors and improve phrasing. Pay particular attention to verb tenses and article usage.

Answer: According to your review, we have corrected grammatical errors and improved phrasing, particularly the verb tenses and article usage.

Minor Comments: - The abstract is very long and detailed. Consider shortening it and focusing on the key objectives, methods, and top-level findings. The term "commodity" is used throughout. While not incorrect, "product" or "brand" is more common in marketing literature for this context (e.g., milk tea).

Answer: According to your review, we have shorted the abstracts and focusing on the key objectives, methods, and top-level findings. "commodity" was replaced with products in manuscript.

Conclusion: - This paper has the potential to be a valuable contribution to the literature on consumer-generated content and advertising. The experimental design is strong, and the research questions are pertinent. However, significant revisions are required to meet the standards of a leading journal. These revisions primarily involve enhancing methodological transparency, providing full data access, improving statistical reporting rigor, and polishing the language.

Answer: Firstly, the reasons for adopting ANOVA have been added before the experimental, and the analysis part has been separated in the experimental.

Secondly, We have uploaded the full data as supporting Information files.

Thirdly, we have revised the paper, especially in verb tenses and article usage.

Thank you and best regards.

---

## [Decision Letter · Decision Letter 1]

23 Dec 2025

A Study of Consumer-Generated Advertising--- An experimental analysis based on purchase experiences, information channels, and privacy costs

PONE-D-25-32677R1

Dear Dr. Cai,

We’re pleased to inform you that your manuscript has been judged scientifically suitable for publication and will be formally accepted for publication once it meets all outstanding technical requirements.

Kind regards,

Reza Rostamzadeh

Academic Editor

PLOS One

Additional Editor Comments (optional):

Reviewers' comments:

Reviewer's Responses to Questions

**Comments to the Author**

Reviewer #1: All comments have been addressed

2. Is the manuscript technically sound, and do the data support the conclusions?

Reviewer #1: Yes

3. Has the statistical analysis been performed appropriately and rigorously?

Reviewer #1: Yes

4. Have the authors made all data underlying the findings in their manuscript fully available?

Reviewer #1: Yes

5. Is the manuscript presented in an intelligible fashion and written in standard English?

Reviewer #1: Yes

Reviewer #1: The revised version of the paper was revised according to the comments that have been raised earlier.

**Do you want your identity to be public for this peer review?** For information about this choice, including consent withdrawal, please see our Privacy Policy

Reviewer #1: **Yes:** Tawfeeq M Alanazi

---

## [Editor Report · Acceptance letter]

PONE-D-25-32677R1

PLOS One

Dear Dr. Cai,

I'm pleased to inform you that your manuscript has been deemed suitable for publication in PLOS One. Congratulations! Your manuscript is now being handed over to our production team.

Kind regards,

on behalf of

Dr. Reza Rostamzadeh

Academic Editor

PLOS One